# Gene Expression and Drug Sensitivity Analysis of Mitochondrial Chaperones Reveals That HSPD1 and TRAP1 Expression Correlates with Sensitivity to Inhibitors of DNA Replication and Mitosis

**DOI:** 10.3390/biology12070988

**Published:** 2023-07-11

**Authors:** Mai Badarni, Shani Gabbay, Moshe Elkabets, Barak Rotblat

**Affiliations:** 1Shraga Segal Department of Microbiology, Immunology and Genetics, Faculty of Health Science, Ben-Gurion University of the Negev, Beer-Sheva 84105, Israel; maibda@post.bgu.ac.il; 2National Institute for Biotechnology in the Negev, Ben-Gurion University of the Negev, Beer-Sheva 84105, Israel; gabbays@post.bgu.ac.il; 3Department of Life Sciences, Faculty of Life Science, Ben-Gurion University of the Negev, Beer-Sheva 84105, Israel

**Keywords:** metabolism, CCLE, co-expression, bioinformatics

## Abstract

**Simple Summary:**

Mitochondria—the energy-producing organelles of a cell—contain numerous enzymes essential for many cellular processes. Enzymes, which are imported into the mitochondria in linear form, must be properly folded to become functional. It is known that the folding is facilitated by molecules known as chaperones. However, it is not known whether the folding of specific mitochondrial proteins (enzymes) is dependent on particular mitochondrial chaperones or whether specific chaperones are linked to drug sensitivity in tumor cells. Here, we analyzed drug sensitivity and gene expression data for cancer cells treated with various drugs and found that levels of certain chaperones correlated with the cells’ response to different drug classes. Additionally, we experimentally confirmed our findings by showing that a certain mitochondrial chaperone could protect tumor cells from one type of drug while making them more sensitive to another. These findings lay down the groundwork for understanding the roles of mitochondrial chaperones in cancer therapeutics, with implications for personalized medicine.

**Abstract:**

Mitochondria—critical metabolic hubs in eukaryotic cells—are involved in a wide range of cellular functions, including differentiation, proliferation, and death. Mitochondria import most of their proteins from the cytosol in a linear form, after which they are folded by mitochondrial chaperones. However, despite extensive research, the extent to which the function of particular chaperones is essential for maintaining specific mitochondrial and cellular functions remains unknown. In particular, it is not known whether mitochondrial chaperones influence the sensitivity to drugs used in the treatment of cancers. By mining gene expression and drug sensitivity data for cancer cell lines from publicly available databases, we identified mitochondrial chaperones whose expression is associated with sensitivity to oncology drugs targeting particular cellular pathways in a cancer-type-dependent manner. Importantly, we found the expression of TRAP1 and HSPD1 to be associated with sensitivity to inhibitors of DNA replication and mitosis. We confirmed experimentally that the expression of HSPD1 is associated with an increased sensitivity of ovarian cancer cells to drugs targeting mitosis and a reduced sensitivity to drugs promoting apoptosis. Taken together, our results support a model in which particular mitochondrial pathways hinge upon specific mitochondrial chaperones and provide the basis for understanding selectivity in mitochondrial chaperone-substrate specificity.

## 1. Introduction

Mitochondria are central metabolic hubs involved in a wide array of cellular pathways, including cell proliferation, growth, differentiation, and programmed cell death. Most mitochondrial proteins are coded in the nuclear genome, translated in the cytosol, and imported into the mitochondria in an unfolded state, after which they are folded by mitochondrial chaperones (reviewed in [1]). There are 15 mitochondrial chaperones, including co-chaperones and proteases, that are responsible for the folding of more than 1000 mitochondrial proteins [2]. However, despite extensive research on protein folding, it is not known whether the folding of specific proteins depends on particular mitochondrial chaperones. Answering this question is essential if we are to better understand those biological processes in which particular mitochondrial proteins play vital roles. The expression and subsequent import of these proteins into the mitochondria may be enhanced in certain cellular processes, e.g., differentiation. In such a case, the availability of particular chaperones needed to fold these key proteins may become a limiting factor.

Metabolic reprogramming is a hallmark of cancer, with mitochondria playing critical roles in various aspects of cancer biology, such as metastasis [3], cell proliferation and apoptosis, and drug resistance [4,5]. These processes are influenced by changes in the expression of the mitochondrial proteins that promote or inhibit specific mitochondrial functions [6]. Therefore, it should be possible to identify mitochondrial proteins involved in resistance to cancer therapy by analyzing gene expression profiles of resistant and sensitive cancer cell lines [7]. Indeed, the association between the expression of mitochondrial chaperones and drug resistance has been described in several studies [8,9,10,11,12]. For example, it was shown that TRAP1, a mitochondrial HSP90 chaperone, supports the expression of multiple mitochondrial enzymes and mitochondrial metabolism in tumor cells [13].

However, the specific mitochondrial chaperones associated with drug resistance in particular cancer types remain to be identified. The resolution of this question will advance our basic understanding of which mitochondrial pathways depend on which chaperones in cancer cells and, at the translational level, how and if mitochondrial chaperone inhibitors/activators can serve as potential drugs in medical oncology.

Here, we analyzed gene expression and drug sensitivity data for multiple cancer cell lines (CCLE and GDSC databases) and found that the expression of specific mitochondrial chaperones is associated with resistance or sensitivity to particular anti-cancer drugs in a tumor-type-dependent manner. The CCLE and GDSC databases were extensively used for studying cancer biology, validating cancer targets, and identifying drug sensitivities in different cancer types. Here, we focused on mitochondrial chaperones and how their expression correlates with drug sensitivity in a cancer-type-specific manner. Our results support a model in which specific mitochondrial chaperones are essential to supporting the activity of specific mitochondrial processes.

## 2. Materials and Methods

### 2.1. Data Mining and Drug-Gene Correlations

Gene expression data were downloaded from the Cancer Cell Line Encyclopedia (CCLE)—Broad DepMap Portal (22Q1) (https://depmap.org/portal/download/, accessed on 20 July 2021). The CCLE provides RNAseq values for 16,383 coding genes across 1377 cancer cell lines. Drug response data were downloaded from the Genomics of Drug Sensitivity in Cancer (GDSC) resource (https://www.cancerrxgene.org, accessed on 20 July 2021). The GDSC provides measurements of the area under the dose-response curve (AUC) at a range of drug concentrations for 449 drugs across 1681 cell lines. We removed from our downloaded file drugs for which there was no assignment to a drug category and those assigned as “Other” and “Unclassified”. Drug categories that contained drugs with similar target pathways were combined, as follows: drugs that target EGFR (ErbB-1) “EGFR signaling” were combined with “ErbBs signaling” drug category, “IGF1R signaling” drugs were combined with “RTK signaling” drugs, and drugs that were assigned as “Other, kinases” were combined with “Kinases”. In addition, we included only drug categories that contained at least 10 drugs. Overall, 16 drug categories remained.

For each drug, we excluded cell lines for which there were no gene expression values and cell lines derived from blood, bone, or soft tissue, as we aimed to focus exclusively on solid tumors. We also excluded tissue subtypes with less than 10 tested cell lines. Next, for each drug–gene pair, we calculated Pearson’s correlation coefficient between basal gene expression and AUC values across cell lines belonging to the same tissue subtype. To account for variations in the number of cell lines available for each drug, we applied Fisher’s z-transformation to the correlation coefficients, as recommended by Rees et al. [14]. Overall, we obtained positive and negative drug–gene correlations for 372 drugs and 16,383 genes across 1179 cell lines. Drug–gene correlations (Z-scores) beyond ±1.7 were considered significant (Appendix A).

### 2.2. Global Score Calculation

For each gene, we counted the number of correlated drugs in each drug category (Z-score beyond ±1.7) across all tissue subtypes, and we calculated, separately, the percentage of significant positive and negative correlations as a global score as (Score_global_) as follows:(1)$${Score}_{global}=\frac{number\;of\;significat\;correlations}{number\;of\;drugs\;in\;each\;category*number\;of\;cancer\;subtypes}\times 100$$

A Z-score beyond ±1.7 was considered significant based on an empirical null distribution obtained from randomized data. To determine which type of cancers contributed most to the Score_global_ for each gene and each drug category, we calculated the percentage of the drugs with a significant Z-score in each cancer subtype.

### 2.3. Code

Correlation analysis was performed using Python programming language (version 3.8.5) within a Windows environment, with aid of ‘pandas’ and ‘numpy’ packages. Data visualization was performed using ‘Matplotlib’ and ‘Seaborn’ packages. The Python code is provided at https://github.com/MaiBda/Mito_chap, accessed on 20 June 2023.

### 2.4. Tumor Cell Lines

Ovarian cancer cell lines were obtained and grown as previously described [15]. The cells were maintained at 37 °C in a humidified atmosphere and 5% CO_2_. IGROV1 and SKOV3 cell lines were cultured in RPMI-1640; OAW42 and Ca-ov3 cell lines were cultured in DMEM. The growth media were supplemented with 10% fetal bovine serum (FBS), 1% l-glutamine 200 mM, and 100 units each of penicillin and streptomycin.

### 2.5. IC_50_ Assay

Cells were seeded in flat-bottom 96-well plates, treated with increasing concentrations of the tested drug (0–10 μM cisplatin), and allowed to proliferate for four days. Cells were then fixed and stained with crystal violet (1 g/L) for 30 min at room temperature. Following rinsing, crystal violet was dissolved out with 10% acetic acid, and absorbance was measured at 570 nm (BioTek Epoch spectrophotometer, Winooski, VT, USA). Half-maximal inhibitory concentration (IC_50_) values were calculated using GraphPad Prism 9 Software.

### 2.6. Western Blotting

Cells were washed three times with phosphate-buffered saline (PBS) and scraped into ice-cold lysis buffer (50 mM HEPES [pH 7.5], 150 mM NaCl, 1 mM EDTA, 1 mM EGTA, 10% glycerol, 1% Triton X-100, 10 μM MgCl_2_). The lysis buffer was supplemented with phosphatase inhibitor cocktails (BiotoolB15001A/B) and protease inhibitor (P2714-1BTL, Sigma-Aldrich, Burlington, MA, USA). The cells in the lysis buffer were then placed on ice for 30 min, followed by 3 min of ultrasonic cell disruption. Lysates were cleared by centrifugation at 14,000 rpm for 10 min at 4 °C. Supernatants were collected, and protein concentration was determined by the Bio-Rad Protein Assay (500-0006, Hercules, CA, USA). Lysates were mixed with LDS sample buffer (4×) (B0007, Thermo Fisher Scientific, Waltham, MA, USA) and NuPAGE Sample Reducing Agent (10×) (NP0009, Thermo Fisher Scientific) and boiled at 95 °C for 5 min. From the total lysate, 20 mg was separated on NuPAGE 12.5% SDS–PAGE and blotted onto PVDF membranes (Bio-Rad Trans blot TurboTM transfer pack, 1704157, Hercules, CA, USA). Membranes were blocked for 1 h in blocking solution [5% BSA (Amresco 0332-TAM) in Tris-buffered saline (TBS) with 0.1% Tween] and then incubated overnight with primary antibodies diluted in blocking solution, supplemented with sodium azide (S2002-5G, Sigma-Aldrich, Burlington, MA, USA). Mouse and rabbit horseradish peroxidase (HRP)-conjugated secondary antibodies were diluted in blocking solution. Protein–antibody complexes were detected by chemiluminescence (Westar Supernova XLS3.0100, Cyanagen, Santa Clara, CA, USA) and Westar Nova 2.0 (XLS071.0250, Cyanagen), and images were captured using the Azure C300 Chemiluminescent Western Blot Imaging System (Azure Biosystems, Dublin, CA, USA).

### 2.7. Cell Viability and Caspase-3 Activity Assay

The CellTiter-Glo^®^ luminescent cell viability assay and Caspase-Glo^®^ 3/7 assay were performed on each sample according to the manufacturer’s instructions (Promega, WI, USA). Briefly, cells were plated in triplicate onto white opaque, flat-bottom 96-well plates. After 24 h of incubation, cells were treated with the indicated drug for the desired time. Cells were then incubated for 10 min with CellTiter-Glo reagent or 2 h with Caspase-Glo reagent, and luminescence was measured. Background luminescence was measured in a medium without cells and subtracted from the experimental values.

### 2.8. Annexin V-FITC Flow Cytometric Analysis

The Dead Cell Apoptosis Kit with annexin V-FITC and propidium iodide (PI) for flow cytometry (Invitrogen, MA, USA) was used to detect apoptotic cells. Briefly, cells were treated with the indicated drug for 24 h. Floating and trypsinized cells were washed in ice-cold PBS with 0.5% BSA and stained using the Dead Cell Apoptosis Kit according to the manufacturer’s instructions. A fluorescence-activated cell sorting (FACS) analysis was performed using FlowJo software v8.8, and 10,000 events were represented as dot plots.

### 2.9. Antibodies

HSPD1 (sc-13115) was purchased from Santa Cruz Biotechnology (Dallas, TX, USA). Anti-Actin (0869100-CF) was purchased from MP Biomedicals (Irvine, CA, USA). Mouse and rabbit horseradish peroxidase (HRP)-conjugated secondary antibodies were purchased from GE Healthcare (Chicago, IL, USA).

### 2.10. Drugs and Reagents

Cisplatin was provided by Soroka medical center, KHS101 was purchased from Sigma-Aldrich (Burlington, MA, USA), and ABT737 was purchased from Selleckchem (Houston, TX, USA). All drugs were dissolved in DMSO.

### 2.11. Statistics

In vitro experiments were repeated 2–3 times. Statistical analysis was performed using GraphPad Prism software, and results are presented as means ± SEM. For comparisons between two groups, *p*-values were calculated using Student’s *t*-test. For comparisons between multiple groups, *p*-values were calculated using one-way ANOVA. *p*-values of 0.05, 0.01, 0.001, and 0.0001 were considered statistically significant, as indicated by *, **, ***, and **** in the figures.

## 3. Results

### 3.1. Identification of Genes Whose Expression Correlates with Drug Sensitivity

We hypothesized that the expression of particular mitochondrial chaperones in tumor cell lines is linked to the response (sensitivity/resistance) to specific drugs in specific cancer types. Using data from the CCLE and GDSC databases, we calculated the correlation between the expression of each mitochondrial chaperone and the sensitivity of cancer cell lines derived from different tumor types towards drugs targeting particular pathways by using the pipeline described below.

We downloaded the mRNA expression levels of 16,383 genes across 1377 cancer cell lines from CCLE and drug sensitivity measurements of 499 drugs across 1681 cancer cell lines from GDSC (Figure 1A). We filtered out cell lines for each drug for which gene expression data were unavailable, and we excluded tissue types with less than 10 tested cell lines (see Section 2 for details). In addition, we excluded cell lines derived from blood, bone, and soft tissue, as we aimed to focus on solid tumors. As a result, we obtained a total dataset composed of 16,383 genes, 372 drugs, and 1179 cell lines.

Next, for each drug–gene pair, we calculated Pearson’s correlation coefficient for the expression level of the gene and the potency of the drug (AUC) across the panel of cell lines. Since the tissue of origin of the cells might play a confounding role in defining drug response [14], for each drug–gene pair, we calculated Pearson’s correlations separately for each sub-tissue in our dataset (Figure 1B) (all sub-tissues are listed in Figure 1D). We used a z-transformed version of Pearson’s correlation coefficient (Z-score) to adjust for variation in cell line number in each sub-tissue [14]. A positive Z-score indicates a resistant phenotype, while a negative Z-score indicates sensitivity. We set a cutoff based on an empirical null distribution obtained from randomized data to identify significant drug–gene associations. Z-scores beyond ±1.7 were considered significant and were used for further analysis (Appendix A). Note that this is a commonly used method that is not intended for detecting individual drug–gene correlations (which would necessitate correction for multiple testing). Rather, much like signature identification in analyzing differential gene expression, the aim is to identify groups of genes that exhibit a (slight) correlation with drug response, as shown by Fernández-Torras et al. [16].

To determine the drug categories that were strongly linked to gene expression, for each gene and drug category, we computed a global correlation score (Score_global_) by calculating, separately, the percentage of positive and negative correlations that were found to be significant (Z-score beyond ±1.7) across all cancer subtypes (Figure 1C). For the resistance Score_global_ (Res-Score_global_), we obtained a range between 0 and ~20% (Appendix A—left panel). In contrast, for the sensitive Score_global_ (Sen-Score_global_), we obtained a range between 0 and ~40% (Appendix A—right panel). We performed a distribution analysis for all positive and negative Score_global_. We set a cutoff at the 5% end of the distribution (Appendix A) to identify the most significant drug–gene category Score_global_. To determine the type of cancer that contributed most to both resistance and sensitive Score_global_, for each drug–gene category, we calculated the percentage of the drugs that had a significant Z-score in each cancer subtype (Figure 1D).

Our next step was to validate our pipeline, i.e., its reliability in identifying a correlation between a clinically tested drug, its target gene, and cancer type. To do this, we used the gene ErbB2, which encodes the human epidermal growth factor receptor (2 HER2). ErbB2 is a well-studied oncogene whose role in cancer development has been well-established, and HER2 inhibitors have been developed and are used successfully in the clinic for treating tumors expressing high levels of HER2 [17]. We, therefore, expected to obtain a significant Sen-Score_global_ between ErbB2 mRNA expression and the drug category that contains drugs that target ErbB2 signaling (which we designated ErbBs sign.). Indeed, the Sen-Score_global_ for ErbB2 gene and the ErbBs sign. drug category was found to be significant (Figure 2A). Consistent with the literature, we found that from all of the cancer sub-tissues, breast cancer contributed the most to the Sen-Score_global_ of ErBb2, as high ErbB2 expression was significantly associated with sensitivity to 62% of the drugs in the ErbBs sign. category in breast cancer (Figure 2B). For non-small-cell lung squamous cell carcinoma (designated Lung_Nsclc_SCC) and gastric (designated Stomach) cancer cell lines, ErbB2 expression was significantly associated with sensitivity to 44% of drugs targeting ErbBs (Figure 2B). In clinical practice, ErbB2 is an established therapeutic target for ErbB2-positive breast cancer [18]. Various agents that target ErbB2, including trastuzumab, pertuzumab, and lapatinib, have been widely used—and shown to be effective—in the treatment of patients with ErbB2-positive breast cancer. Recently, anti-ErbB2 antibody–drug conjugates, such as trastuzumab, were approved for ErbB2-positive gastric cancers, and the addition of trastuzumab to first-line chemotherapy has improved the overall survival of ErbB2-positive gastric cancer patients, becoming the standard-of-care treatment for this group of patients [19,20]. These findings confirm our pipeline’s credibility in identifying genes that are associated with drug sensitivity in specific cancer tissue types where such associations exist.

Importantly, our analysis also showed that a high expression of ErbB2 is associated with resistance to drugs targeting protein stability and degradation (designated the Protein stab. & deg.) (Figure 2A). We found that colon (designated Large_intestine) and non-small-cell lung adenocarcinoma (designated Lung_Nsclc_AC) contributed the most to the Res-Score_global_ of the Protein stab. & deg. drug category, as high ErbB2 expression in those cancer cell lines was found to be significantly associated with resistance to 40% of the drugs targeting protein stability and degradation, followed by breast cancer in which 30% of the drugs targeting Protein stab. & deg. were associated with resistance (Figure 2C). We also found that high expression of ErbB2 was associated with resistance to drugs that target ERK/MAPK signaling (designated ERK/MAPK sign) (Figure 2A), mainly in breast, Lung_Nsclc_AC, and thyroid cells, as these cell lines were associated with resistance to 57%, 30% and 29% of MAPK-pathway-targeting drugs, respectively (Figure 2C).

Given that various drugs target distinct proteins within each drug category, we sought to investigate whether drugs targeting similar proteins exhibit comparable cellular response profiles. In addition, we asked how different cancer subtypes respond to these drugs. To this end, a hierarchical clustering of significant drug categories across all cancer subtypes was used to group cancers with similar response patterns to a particular drug. Clustering of the ErbBs sign. drug category revealed that drugs with similar targets tend to be clustered together, and drugs targeting ErbB2, 3, and 4 were more potent in cancer cell lines with high ErbB2 expression as compared with drugs targeting EGFR (ErbB1) alone (Figure 2D). As we anticipated, the cancer subtypes with the highest percentage of significant correlations with drugs (breast, Lung_Nsclc_SCC, and stomach) were found closer together (Figure 2D). Similarly, in the Protein stab. & deg. category, clustering analysis showed that drugs with the same target tend to be clustered together, and cell lines with high ErbB2 expression were less responsive to these classes of drugs in general and to inhibitors of a chaperone—heat shock protein 90 (HSP90)—in particular (Figure 2E).

### 3.2. Tumor Necrosis Factor Receptor-Associated Protein 1 (TRAP1) Expression Facilitates Sensitivity to Chemotherapy

To explore whether and how the expression of mitochondrial chaperones affects drug response, we applied our pipeline to a list of 15 mitochondrial chaperones and co-chaperones (Appendix A) [2]. Nine chaperones and co-chaperones showed a significant positive or negative Score_global_ in one or more drug categories (Figure 3 and Figure 4, Appendix A). Among the chaperones that had the highest number of significant associations with different drug categories was tumor necrosis factor receptor-associated protein 1 (TRAP1). We observed that a high expression of TRAP1 is associated with an increased sensitivity to drugs that target metabolism, DNA replication (designated DNA rep.), and Mitosis (Figure 3A). Interestingly, several drugs in the DNA rep. and Mitosis categories are chemotherapy agents that have been approved and widely used for the treatment of cancer patients, such as docetaxel and paclitaxel, etoposide, cisplatin, and doxorubicin. The Sen-Score_global_ in the DNA rep. and Mitosis classes was found in cell lines originating from ovarian and kidney cancer. In ovarian cancer cell lines, high TRAP1 expression was significantly associated with sensitivity to 54% of DNA rep. inhibitors and 42% of Mitosis inhibitors, while in kidney cancer cell lines, it was significantly associated with sensitivity to 64% of DNA rep. inhibitors and 37% of Mitosis inhibitors (Figure 3B).

The hierarchical clustering of the cancer subtypes across DNA rep. drugs indicates that kidney and ovarian cancer subtypes are clustered together and display the strongest and most significant associations between high TRAP1 expression and sensitivity to DNA rep. drugs (Figure 3D). However, the clustering of the drugs suggests that this relationship was not specific to any particular drug target, as significant correlations were found across all drug targets in these cancer subtypes (Figure 3D). The results also indicated that high TRAP1 expression is associated with resistance to ErbBs sign. drugs (Figure 3A), particularly in stomach cancer, where TRAP1 expression is significantly correlated with resistance to 44% of the drugs in this category (Figure 3C). Hierarchical clustering of the cancer subtypes and drugs showed that stomach cancer was the most resistant to drugs targeting both ErbBs and EFGR (Figure 3E).

### 3.3. Heat Shock Protein Family D Member 1 (HSPD1) Expression Is Correlated with Sensitivity to Chemotherapy

HSPD1 was the chaperone with the most significant associations with different drug categories, including Protein stab. & deg., chromatin acetylation (designated Chromatin acet.), and ERK/MAPK sign. Similar to our findings for TRAP1, high expression levels of HSPD1 were significantly associated with sensitivity to drugs that target DNA rep. and Mitosis (Figure 4A). We noticed that cell lines originating from ovarian and kidney cancer contributed the most to the Sen-Score_global_ for these drug classes, as in ovarian cancer cell lines, high HSPD1 expression was significantly associated with sensitivity to 39% of DNA rep. inhibitors and 42% of Mitosis inhibitors. In kidney cancer cell lines, high HSPD1 expression was significantly associated with sensitivity to 30% of DNA rep. inhibitors and 32% of Mitosis inhibitors (Figure 4B). Hierarchical clustering indicated that kidney and ovarian cancer subtypes were clustered together and displayed the strongest and most significant associations between high HSPD1 expression and sensitivity to DNA rep. drugs, but no specific response patterns were shown for drugs with similar targets (Figure 4D).

### 3.4. HSPD1 Expression Facilitates Sensitivity to Cisplatin and the Apoptosis Inducer, ABT737

To validate the results of our pipeline in culture, we used ovarian cancer cell lines to explore whether a high level of HSPD1 is indeed associated with sensitivity to chemotherapy. Specifically, we first measured the expression levels of HSPD1 in four ovarian cancer cell lines using Western blot analysis (Figure 5A). Thereafter, we treated these ovarian cancer cell lines with increasing concentrations of the chemotherapy drug cisplatin, determined cell viability, and calculated IC_50_ (half maximal inhibitory concentration) values. We found that cell lines with higher HSPD1 expression (IGROV1 and Ca-OV3) were sensitive to cisplatin and exhibited significantly lower viability (Figure 5B—left panel) and significantly lower IC_50_ values compared to the cell lines with lower expression levels of HSPD1 (Figure 5B—right panel). These results suggest that high HSPD1 expression may be a biomarker of sensitivity to certain chemotherapy drugs in ovarian cancer.

Our model argues that specific chaperones cater to the folding of specific proteins; therefore, their expression will be correlated with sensitivity or resistance to specific drugs. We found that HSPD1 expression is associated with sensitivity to cisplatin, and we asked whether HSPD1 expression is associated with resistance to another class of drugs. According to our analysis, HSPD1 expression was not found to be significantly associated with sensitivity to apoptosis-inducing drugs, and we chose ABT373 as a test case. We performed cell-based assays to measure the apoptosis rate of tumor cells following treatment with different concentrations of the apoptosis inducer ABT737 (the HSPD1 inhibitor), KHS101, or a combination of the two. We determined cell viability and caspase-3 activity using the CellTiter-Glo luminescent cell viability and Caspase-Glo 3/7 assays, respectively. The ratio between dead and viable cells in IGROV1 and OAW42 cell lines showed that treatment with different concentrations of ABT737 or KHS101 alone slightly increased apoptosis. However, a significant increase in cell death was observed when the two agents were combined (Figure 5C,E). We confirmed these results using annexin V staining to detect apoptotic cells following the treatment of IGROV1 and OAW42 tumor cells with either ABT737 or KHS101 alone or a combination of the two agents. We found that the treatment of the tumor cells with the ABT737/KHS101 combination increased apoptosis compared to treatment with a single agent (Figure 5D,F). These results show that HSPD1 expression is not associated with sensitivity to all kinds of cell-death-inducing drugs.

### 3.5. HSPD1 and TRAP1 Expression Is Correlated with Sensitivity to Similar Drug Categories

We then sought to determine whether the expression of particular mitochondrial chaperones is correlated with sensitivity or resistance to similar drug categories by using Score_global_ and hierarchical clustering. We found that HSPD1 and TRAP1 clustered together and are associated with sensitivity to drugs targeting protein stability and folding, DNA replication, and Mitosis (Figure 6A). However, an examination of drug resistance showed HSPD1 and TRAP1 are in separate clusters (Figure 6B), suggesting a dissimilarity in their functional characteristics related to drug resistance. Drug categories that were clustered together, such as DNA rep., Mitosis, and Protein stab. & deg., were mostly correlated with the expression of mitochondrial chaperones in terms of sensitivity (Figure 6A). In contrast, Chromatin acet. and ErbB2 sign. drug categories were mostly correlated with the expression of mitochondrial chaperones in terms of resistance (Figure 6B).

The expression of CLPP, SPG7, HTRA2, and AFG3L2 were associated with sensitivity to various drug categories. In particular, CLPP and SPG7 cluster together under Chromatin acet. and ErbBs sign. categories. CLPP is an interesting drug target in cancer as inhibitors and activators have been identified and are candidate cancer drugs [21]. Less is known about SPG7 as a drug target in cancer. Interestingly while AFG3L2 interacts with SPG7 in the mitochondria [22], its expression is correlated with different drug categories.

## 4. Discussion

Cells adjust their metabolism to support changes in biological states and nutrient availability. This principle is exemplified in tumor cells, as they consume less ATP than the normal tissue from which they are derived [23]. In tumor cells, the expression levels of metabolic enzymes in the cytosol and the mitochondria are altered to exploit metabolic pathways to overcome therapeutic targeting [24]. Since proteins are imported into the mitochondria in an unfolded state and must fold to become functional, it is important to understand which enzymes or pathways depend on certain chaperones for their folding. Such information can infer the outcome of targeting a specific chaperone in a specific tumor type. In this study, we were surprised to find that the expression of mitochondrial chaperones was linked more strongly to drug sensitivity than drug resistance (Figure 6). The mitochondrial chaperones with the strongest association with drug sensitivity were HSPD1 and TRAP1.

TRAP1, a well-known member of the Hsp90 chaperone family, is a key factor in maintaining mitochondrial stability and intracellular balance [25]. Our results show that high TRAP1 expression was significantly correlated with a favorable response to chemotherapy (Figure 3). This was particularly true in ovarian cancer, with a considerable impact on overall survival [26]. Conversely, low TRAP1 expression was linked to cisplatin resistance in ovarian cancer cells [27] and poor outcomes in cervical, bladder, and clear renal cell carcinoma [25]. On the other hand, we found that high TRAP1 expression was associated with resistance to inhibitors of ErbBs’ signaling (Figure 3E). In agreement with our findings, it was shown that TRAP1 is a determinant of metabolic rewiring in colorectal cancer and favors resistance to the EGFR inhibitor, cetuximab [28].

The role of TRAP1 in tumor development is not well understood. It has been posited that TRAP1 has pro-oncogenic functions based on its increased presence in several types of cancer, such as colorectal, breast, prostate, nasopharyngeal, and lung carcinomas [29]. However, levels of TRAP1 expression also suggest an onco-suppressive role, as its expression levels are decreased in certain types of tumors, such as ovarian, renal, and cervical carcinomas. In addition, in these cases, TRAP1 levels are reduced during tumor progression [25,29,30]. Functionally, TRAP1 was shown to directly promote the activity of the electron transport chain to facilitate cellular respiration in low-nutrient conditions [31] and to promote MYC-dependent tumor cell migration [32]. These results suggest that the expression of TRAP1 may play a role in determining the response of different cancer subtypes to chemotherapy and other targeted therapies.

HSPD1 is an ATP-dependent mitochondrial chaperone, playing an essential role in guaranteeing the correct folding of proteins imported into mitochondria [31]. HSPD1 is overexpressed in many cancer types, and its expression has been correlated with the metastatic potential of cancers and the overall survival of cancer patients [33]. The molecular mechanisms underlying the chemosensitivity of tumor cells expressing high HSPD1 levels, as shown here, remain unclear and require further investigation. HSPD1 was shown to promote oxidative phosphorylation (OXPHOS) in ovarian cancer, as HSPD1 knockdown disrupted mitochondrial integrity and suppressed OXPHOS [34]. One possible explanation for the correlation of high HSPD1 expression with increased sensitivity to mitosis-targeting drugs can be attributed to the fact that epithelial ovarian cancer is highly dependent on OXPHOS to maintain tumor growth and progression [35], and it was reported that high-OXPHOS ovarian cancer cell lines exhibit an increased response to conventional chemotherapies [35]. Thus, cells with higher HSPD1 expression respond better to chemotherapy. In our previous work, we revealed that HSPD1 and TRAP1 exhibit co-expression with a similar set of mitochondrial proteins [36]. This observation may clarify why they are linked to susceptibility to drugs that target similar pathways.

While it was surprising that the expression of the chaperone, HSPD1, was correlated with sensitivity to specific drug categories, it was also surprising that it was not significantly associated with resistance, except towards inhibitors of WNT signaling. In testing whether high HSPD1 expression sensitizes ovarian cancer cells to cisplatin, a drug targeting proliferation, we found that cells with higher HSPD1 expression were indeed more sensitive. To rule out the possibility that HSPD1 sensitizes cells to all types of death-inducing compounds, we tested whether HSPD1 expression sensitizes cells to the BCL2 inhibitor and apoptosis-inducing drug ABT737 and found that HSPD1 inhibitor synergized with the apoptosis inhibitor. These results demonstrated that HSPD1 does not universally sensitize tumor cells to all types of cell-death-inducing drugs but rather may sensitize them to some and protect against others. Note, our analysis did not find a significant correlation between HSPD1 expression and resistance to the apoptosis regulation (designated as Apoptosis reg.) drug category; however, the Score_global_ was very close to the significance cutoff (Figure 4A).

Finally, we relate to a surprising finding from our analysis regarding HSPD1 and its co-chaperone, HSPE1, the human homolog of bacterial GroEL and GroES. These two molecules form a football-like complex that functions to fold substrates [37,38]. Our analysis revealed that the expression of HSPD1 and HSPE1 were associated differently with sensitivity and resistance to different drugs, apart from sensitivity to protein stability and degradation (Figure 6). These results suggest that HSPD1 and HSPE1 may have independent functions, in addition to their joint function, in agreement with the finding that they are co-expressed with different groups of mitochondrial proteins [36].

### Limitations of the Analysis

The correlation analysis of gene expression and drug sensitivity measurements is a valuable tool that can help identify possible links between particular genes and drug responsiveness. Nevertheless, it has some limitations that must be considered, such as the fact that correlation does not imply causation and a correlation between gene expression and drug sensitivity does not necessarily mean that gene expression causes drug sensitivity. Additionally, it remains to be determined whether immortalized cells can accurately represent patient samples [39]. Despite these limitations, our analysis can offer an initial insight into drug response patterns based on gene expression across various cancer types. However, further validation and investigation are required to confirm these findings.

## 5. Conclusions

Our analysis mapped the landscape of mitochondrial chaperone expression and drug sensitivity in cancer cell lines originating from different tumor types. These findings can be used to investigate which particular mitochondrial pathways are supported by specific mitochondrial chaperones. Moreover, our results create a reference catalog linking chaperone expression with drug resistance or sensitivity in particular tumor types. Importantly, this study revealed that the expression of HSPD1 and TRAP1, both potential targets for cancer drugs, is linked to sensitivity to certain drug categories in specific tumor types. This information can be used in personalized medicine to prioritize the use of these inhibitors in particular cancer patients.

## Figures and Tables

**Figure 1 biology-12-00988-f001:**
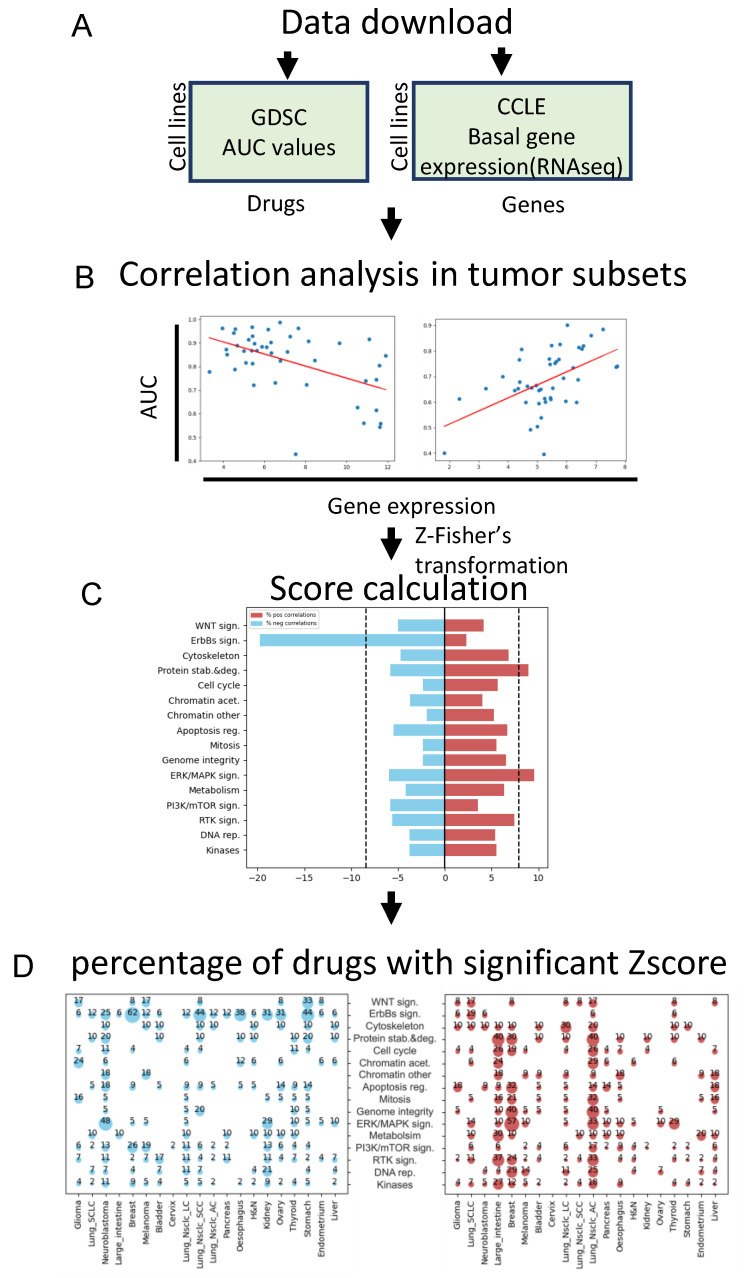
The study outline. (**A**) AUC values and gene expression were downloaded from GDSC and CCLE, respectively. (**B**) Calculation of z-transformed Pearson’s correlation (Z-score) between gene expression and drug sensitivity data for each drug–gene pair. (**C**) Calculation of Sen-Score_global_ and Res-Score_global_ across all drug categories. (**D**) Calculating the percentage of drugs with significant Z-score in each cancer subtype and drug category.

**Figure 2 biology-12-00988-f002:**
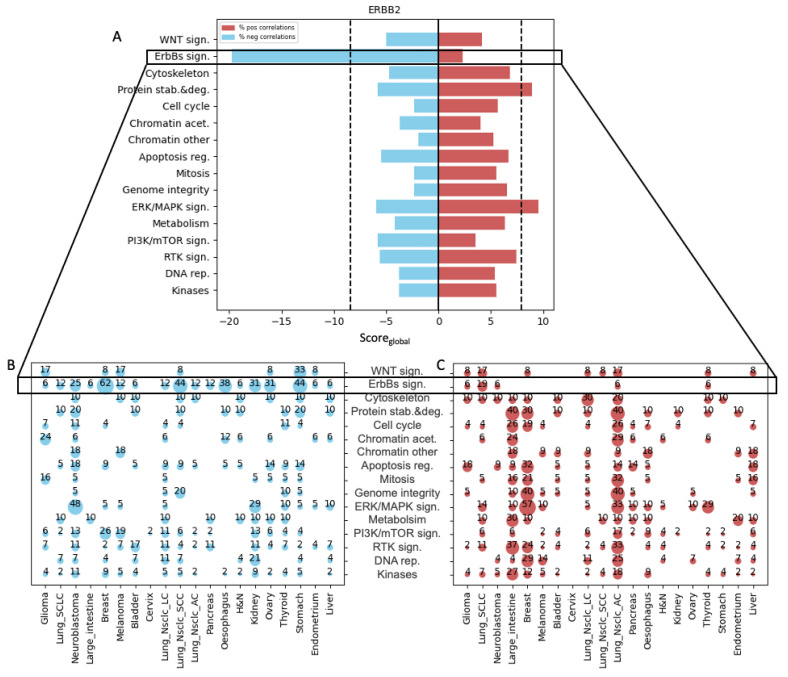
ErbB2 expression is associated with sensitivity to ErbBs sign. inhibitors. (**A**) Bi-directional bar chart showing the Score_global_ of ErbB2 in all drug categories. The blue bar denotes Sen-Score_global_; the red bar denotes Res-Score_global_. The black dotted line indicates the cutoff for the significant score. (**B**) Bubble plot showing the percentage of drugs with significant negative Z-scores in each drug category across all cancer subtypes. (**C**) Bubble plot showing the percentage of drugs with significant positive Z-scores in each drug category across all cancer subtypes. (**D**) Hierarchical clustering of Z-score values between ErbB2 expression and AUC values of ErbBs sign. drugs across all cancer subtypes. (**E**) Hierarchical clustering of Z-score values between ErbB2 expression and AUC values of Protein stab. & deg. drugs across all cancer subtypes.

**Figure 3 biology-12-00988-f003:**
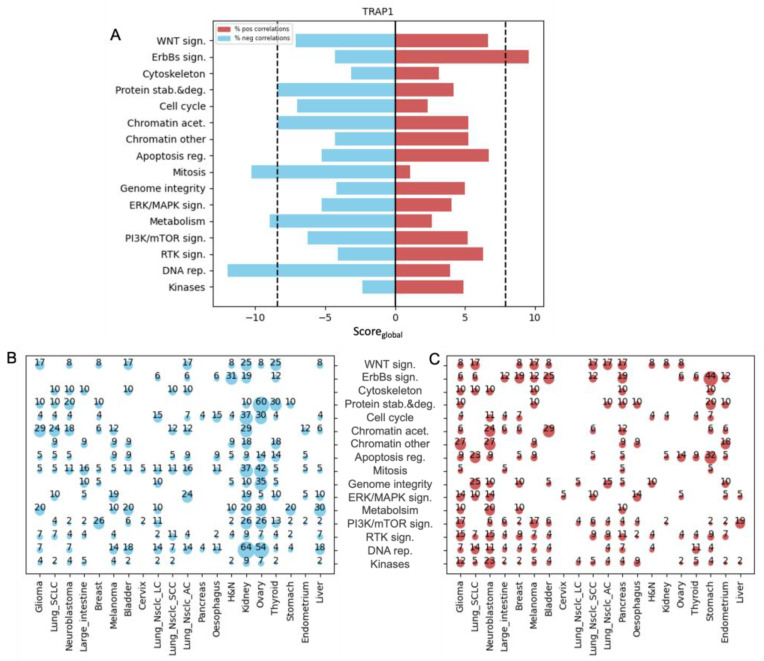
TRAP1 expression is associated with sensitivity to DNA rep. and Mitosis drugs. (**A**) Bi-directional bar chart showing the Score_global_ of TRAP1 in all drug category. The blue bar denotes Sen-Score_global_; the red bar denotes Res-Score_global_. The black dotted line indicates the cutoff for significant scores. (**B**) Bubble blot showing the percentage of drugs with significant negative Z-scores in each drug category across all cancer subtypes. (**C**) Bubble blot showing the percentage of drugs with significant positive Z-scores in each drug category across all cancer subtypes. (**D**) Hierarchical clustering of Z-score values between TRAP1 expression and AUV values of DNA rep. drugs across all cancer subtypes. (**E**) Hierarchical clustering of Z-score values between TRAP1 expression and AUV values of ErbBs sign. drugs across all cancer subtypes.

**Figure 4 biology-12-00988-f004:**
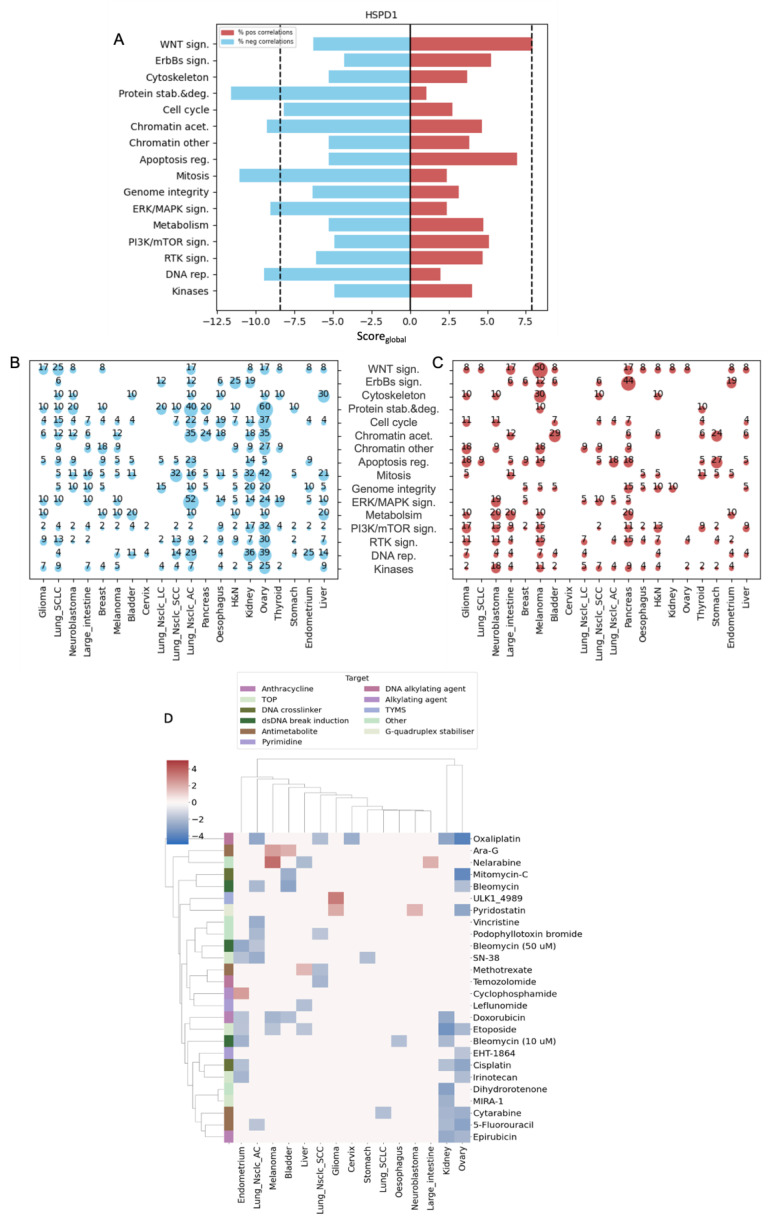
HSPD1 expression is associated with sensitivity to DNA rep. inhibitors. (**A**) Bi-directional bar chart showing the Score_global_ of HSPD1 in all drug categories. The blue bar denotes Sen-Score_global_; the red bar denotes Res-Score_global_. The black dotted line indicates the cutoff for significant scores. (**B**) Bubble plot showing the percentage of drugs with significant negative Z-score in each drug category across all cancer subtypes. (**C**) Bubble plot showing the percentage of drugs with significant positive Z-score in each drug category across all cancer subtypes. (**D**) Hierarchical clustering of Z-score values between HSPD1 expression and AUC values of DNA rep. drugs across all cancer subtypes.

**Figure 5 biology-12-00988-f005:**
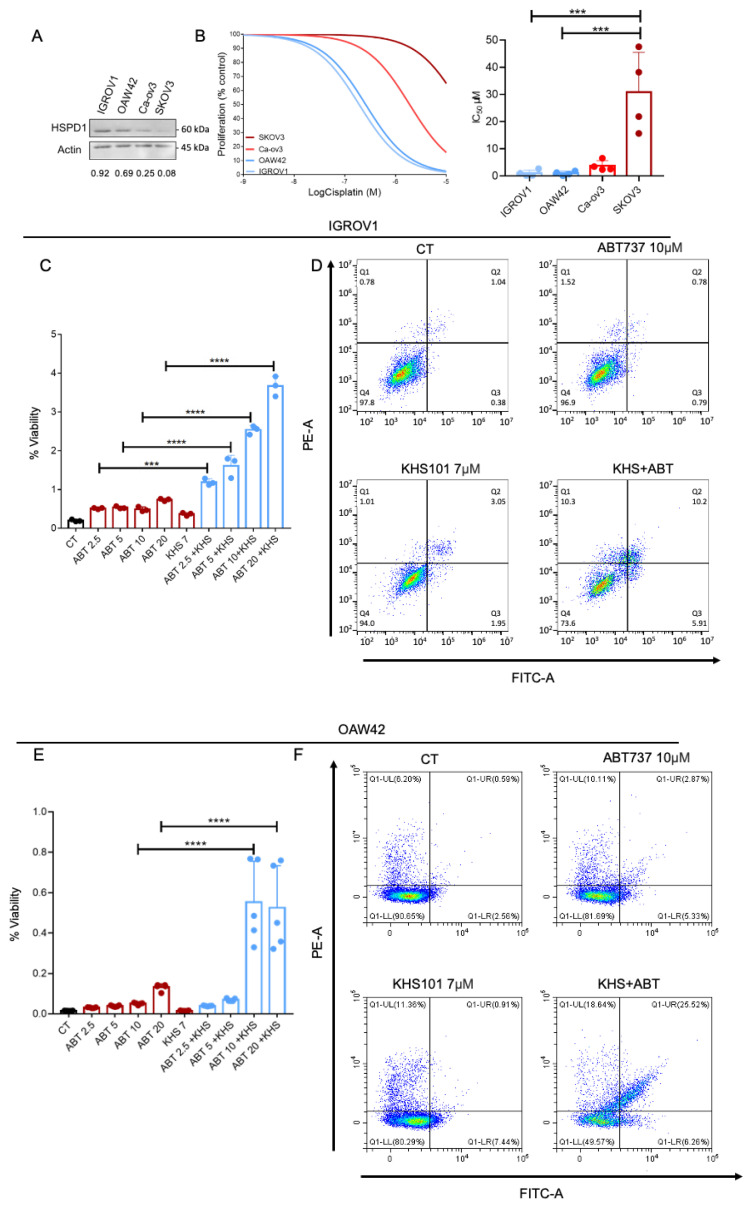
HSPD1 expression protects ovarian cells from apoptosis inducers and DNA rep. inhibitors. (**A**) Western blot analysis showing HSPD1 expression in ovarian cancer cells. Quantification of the bands was done using ImageJ. HSPD1/Actin ratio was calculated and presented in the figure. (**B**) Left panel—Dose-response curve of ovarian cell lines following 4 days of treatment with increasing concentrations of cisplatin. Right panel—IC_50_ values of four separate experiments are presented as means ± SEM. Biological replicates from separate experiments are shown as gray dots. (**C**) IGROV1 cells were treated with the indicated drug. The CellTiter-Glo luminescent cell viability and Caspase-Glo 3/7 assays were performed on each sample according to the manufacturer’s instructions. The ratio between dead and live cells was calculated. The results of a representative experiment are presented as means ± SEM. Biological replicates from separate experiments are shown as dots. (**D**) IGROV1 cells were treated with the indicated drugs and stained with annexin V and PI. Flow cytometry was used to detect apoptotic cells. Representative scatter plots of PI (*y*-axis) vs. annexin V (*x*-axis) are presented. (**E**) OAW42 cells were treated with the indicated drug. The CellTiter-Glo luminescent cell viability and Caspase-Glo 3/7 assays were performed on each sample according to the manufacturer’s instructions. The ratio between dead and live cells was calculated. The results of a representative experiment are presented as means ± SEM. Biological replicates from separate experiments are shown as dots. (**F**) OAW42 cells were treated with the indicated drugs and stained with annexin V and PI. Flow cytometry was used to detect apoptotic cells. Representative scatter plots of PI (*y*-axis) vs. annexin V (*x*-axis) are presented one-way ANOVA *p*-value is shown. *** *p* < 0.001, **** *p* < 0.0001.

**Figure 6 biology-12-00988-f006:**
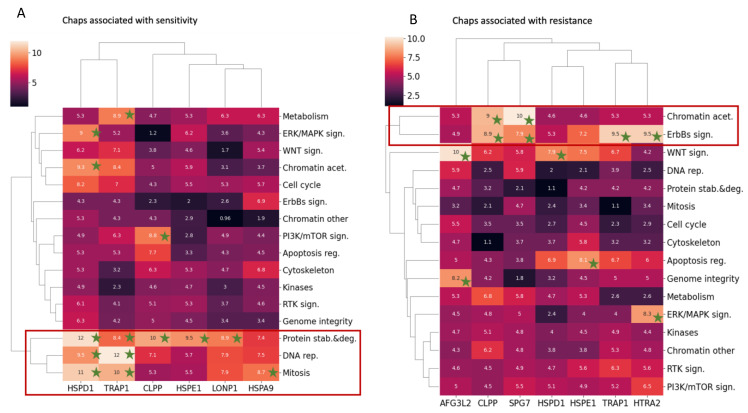
HSPD1 and TRAP1 expression is correlated with sensitivity to similar drug categories. Hierarchical clustering of Score_global_ values of chaperones associated with sensitivity (**A**) and resistance (**B**). Green stars indicate a significant Score_global_. Chaps—chaperones.

## Data Availability

The data presented in this study are available in this article (and Appendix A).

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
