# Peer review of "Gene Expression and Drug Sensitivity Analysis of Mitochondrial Chaperones Reveals That HSPD1 and TRAP1 Expression Correlates with Sensitivity to Inhibitors of DNA Replication and Mitosis"

_biology, 2023, doi:10.3390/biology12070988_

Round 1

Reviewer 1 Report (Previous Reviewer 1)

The authors have addressed my concerns.

Reviewer 2 Report (Previous Reviewer 2)

No amendments requested 

Reviewer 3 Report (Previous Reviewer 4)

This is a very good correlative study establishing a link between the mitochondrial chaperon/cochaperone's expression level and the sensitivity of drugs in different cancer types. In this manuscript (MS), Badarni et al. performed correlation analyses between the expression level of several genes and different drug categories in many cancer cell lines and tissue subtypes. The authors developed a pipeline to calculate Z-scores and Score-global for several gene-drug combinations in many cell lines. The authors analyzed 15 mitochondrial chaperons/cochaperones and found that the higher expression of HSPD1 and TRAP1 significantly correlates with sensitivity to replication and mitosis-targeting drugs. This manuscript is well written, the results are discussed clearly, and the findings support the conclusion. This manuscript is a revised version. Therefore, I expected to see the P by P answers to my comments on the previous version. The P by P answer was not provided to me. Also, the track changes draft was not available to me.

The critical experiment of this MS is the validation of HSPD1 expression and drug sensitivity experimentally. I also expected that the authors would validate the TRAP1 expression and drug sensitivity. Or, the authors would have at least tried to address this part. This was also one of my comments in the previous version. However, I do not see anything about TRAP1 validation in the version provided to me.

Despite this concern, this MS is excellent work and falls within the journal's scope. I fully support the publication of this work and would like to congratulate the authors.

This manuscript is a resubmission of an earlier submission. The following is a list of the peer review reports and author responses from that submission.

Round 1

Reviewer 1 Report

The authors have developed a pipeline for the identification of gene expression-drug correlation. To do so, they have merged data from the CCLE and the GDSC databases. The purpose of this effort was the identification of biomarkers of drug resistance and sensitivity in solid tumours. The robustness of the model was successfully validated using a well-characterized drug target, Her2. Then, the authors apply the methodology to identify mitochondrial chaperones whose expression correlated to sensitivity or resistance to different drug categories. Overall, 9 out 15 chaperones were somehow associated with cancer cells’ response to drug treatment. The authors focused on TRAP1 and HSPD1 as predictor of response to traditional chemotherapeutic drugs.

The paper is well-written and the results are presented in a very clear layout. The idea of looking at mitochondrial chaperone as predictor of drug resistance is quite interesting, however, it might be relevant to clarify how the approach used by the authors differs from what has been done before in terms on gene-drug correlation studies.

Despite their explanation, it is unclear to this reviewer why the authors have tested apoptotic drugs and why ABT737 in particular, as according to their interpretation cisplatin is inducing cell cycle arrest? What are the authors try to assess here?

It is interesting the association between Ovary and Kidney Cancer. Any chance the authors could speculate as to why they tend to cluster together? Could they also clarify the subtype of kidney cancer(s) they have looked at?

Its seems that HSPD1 expression is associated with increased sensitivity to mitotic inhibitors. Is paclitaxel among the tested drugs? Could the authors look, as paclitaxel is used in ovarian cancer and could be interesting to see any correlation that could be used as a biomarker of drug sensitivity.

Minor points

The bubble graphs are often difficult to read. Is there a chance they could be increased in size?

Author Response

The authors have developed a pipeline for the identification of gene expression-drug correlation. To do so, they have merged data from the CCLE and the GDSC databases. The purpose of this effort was the identification of biomarkers of drug resistance and sensitivity in solid tumors. The robustness of the model was successfully validated using a well-characterized drug target, Her2. Then, the authors apply the methodology to identify mitochondrial chaperones whose expression correlated to sensitivity or resistance to different drug categories. Overall, 9 out 15 chaperones

P.O. Box 653, Beersheva 8410501 ISRAEL

Telephone: +972 (0)8 642-8806 = Email: rotblat@bgu.ac.il = Twitter: @BarakRotblat http:// https://barakrotblat.wixsite.com/rotblatlab

were somehow associated with cancer cells’ response to drug treatment. The authors focused on TRAP1 and HSPD1 as predictors of response to traditional chemotherapeutic drugs.

  1. The paper is well-written, and the results are presented in a very clear layout. The idea of looking at mitochondrial chaperone as a predictor of drug resistance is quite interesting, however, it might be relevant to clarify how the approach used by the authors differs from what has been done before in terms on gene-drug correlation studies.

Answer: To clarify how our analysis differs from previous studies, we added this explanation to the introduction:

"The CCLE and GDSC databases were extensively used for studying cancer biology, validating cancer targets, and identifying drug sensitivities in different cancer types. Here, we focused on mitochondrial chaperones and how their expression correlates with drug sensitivity in a cancer- type-specific manner. "

  1. Despite their explanation, it is unclear to this reviewer why the authors have tested apoptotic drugs and why ABT737 in particular, as according to their interpretation cisplatin is inducing cell cycle arrest?

Answer: We thank the reviewer for asking to clarify this important point. In the cisplatin experiment, we measured viability and not cell cycle. We corrected the text in accord.

What are the authors try to assess here?

One possible explanation to the finding that high HSPD1 expression is correlated with increased sensitivity is that HSPD1 is induced in cells that are stressed and, therefore, more sensitive in general. Because our model argues that specific chaperones are associated with response to specific compounds, we asked if HSPD1 can also protect cells against death inducers and chose apoptosis inducers and ABT737 as a test case. Our results show that HSPD1 inhibition synthesized cells to the apoptosis-inducing agent supporting the model where HSPD1 is associated with increased sensitivity to one class of drugs and increased resistance to another. We added text to clarify this point.

  1. It is interesting the association between Ovary and Kidney Any chance the authors could speculate as to why they tend to cluster together? Could they also clarify the subtype of kidney cancer(s) they have looked at?

Answer: Indeed, it is interesting that kidney can ovary cancers cluster together. However, we don’t know why this is the case.

Here is a list of all kidney cancer cell lines used in the analysis and their sub-type:

Cell line

Name

Model ID

COSMIC

ID

TCGA

Classfication

Tissue sub-

type

Disease

BB65-RCC

SIDM001

91

753533

KIRC

kidney

Renal cell carcinoma

HA7-RCC

SIDM001

88

753558

KIRC

kidney

Renal cell carcinoma

LB1047-

RCC

SIDM001

87

753577

KIRC

kidney

Renal cell carcinoma

LB2241-

RCC

SIDM001

86

753578

KIRC

kidney

Renal cell carcinoma

LB996-RCC

SIDM001

80

753585

KIRC

kidney

Renal cell carcinoma

786-0

SIDM001

25

905947

KIRC

kidney

Renal cell carcinoma

A498

SIDM001

24

905948

KIRC

kidney

Renal cell carcinoma

ACHN

SIDM001

23

905950

KIRC

kidney

Papillary renal cell

carcinoma

CAKI-1

SIDM009

41

905963

KIRC

kidney

Renal cell carcinoma

RXF393

SIDM000

86

905978

KIRC

kidney

Renal cell carcinoma

SN12C

SIDM000

94

905979

KIRC

kidney

Renal cell carcinoma

TK10

SIDM001

13

905980

KIRC

kidney

Clear cell renal cell

carcinoma

U031

SIDM001

12

905981

KIRC

kidney

Renal cell carcinoma

G-401

SIDM008

56

907299

UNCLASSIFIED

kidney

Rhabdoid tumor of the

kidney

OS-RC-2

SIDM002

39

909250

KIRC

kidney

Clear cell renal cell

carcinoma

VMRC-

RCZ

SIDM003

17

909781

KIRC

kidney

Renal cell carcinoma

RCC10RGB

SIDM002

35

909974

KIRC

kidney

Renal cell carcinoma

BFTC-909

SIDM009

88

910698

KIRC

kidney

Papillary renal cell

carcinoma

A704

SIDM008

49

910920

KIRC

kidney

Renal cell carcinoma

769-P

SIDM008

03

910922

KIRC

kidney

Renal cell carcinoma

CAL-54

SIDM009

32

910952

KIRC

kidney

Renal cell carcinoma

SW156

SIDM011

62

1240220

KIRC

kidney

Renal cell carcinoma

VMRC-

RCW

SIDM003

18

1240224

KIRC

kidney

Renal cell carcinoma

KMRC-1

SIDM006

11

1298168

KIRC

kidney

Clear cell renal cell

carcinoma

KMRC-20

SIDM006

09

1298169

KIRC

kidney

Clear cell renal cell

carcinoma

NCC010

SIDM002

31

1509073

KIRC

kidney

Renal cell carcinoma

NCC021

SIDM002

32

1509074

KIRC

kidney

Renal cell carcinoma

RCC-FG2

SIDM008

19

1524414

KIRC

kidney

Clear cell renal cell

carcinoma

RCC-JF

SIDM008

18

1524415

KIRC

kidney

Clear cell renal cell

carcinoma

RCC-JW

SIDM008

17

1524416

KIRC

kidney

Clear cell renal cell

carcinoma

RCC-ER

SIDM008

20

1524417

KIRC

kidney

Clear cell renal cell

carcinoma

RCC-AB

SIDM008

21

1524418

KIRC

kidney

Clear cell renal cell

carcinoma

RCC-MF

SIDM008

16

1524419

KIRC

kidney

Clear cell renal cell

carcinoma

  1. Its seems that HSPD1 expression is associated with increased sensitivity to mitotic inhibitors. Is paclitaxel among the tested drugs? Could the authors look, as paclitaxel is used in ovarian cancer and could be interesting to see any correlation that could be used as a biomarker of drug

Answer: Paclitaxel is indeed interesting, and it was included in the tested drugs. Like cisplatin, HSPD1 expression is associated with increased sensitivity to paclitaxel in ovarian cancer, as shown in the hierarchical clustering of Z-score values between HSPD1 expression and AUC values of Mitosis drugs across all cancer subtypes.

Minor points

  1. The bubble graphs are often difficult to Is there a chance they could be increased in size?

Answer: Thank you, corrected.

Reviewer 2 Report

This is a well-conceived, executed, and written study. Comments are minor, few, and superficial. Congratulations to the authors for their good work.

Line 124: specific culture medium used should be specified.

Throughout it should be specified if 96 well plates used are flat- or round-bottom.

Line 147: extra space between bracket and Amresco.

It would be good if figures were shown immediately following the first paragraph of reference within the results section. This would improve readability as there is quite a bit of text separating figures from where they are mentioned. A bit of jumping about is needed. If this is a journal-imposed formatting requirement then please ignore.

Author Response

This is a well-conceived, executed, and written study. Comments are minor, few, and superficial. Congratulations to the authors for their good work.

  1. Line 124: specific culture medium used should be

Answer: Thank you, corrected.

  1. Throughout it should be specified if 96 well plates used are flat- or round-

Answer: Thank you, corrected.

  1. Line 147: extra space between bracket and

Answer: Thank you, corrected.

  1. It would be good if figures were shown immediately following the first paragraph of reference within the results This would improve readability as there is quite a bit of text separating figures from where they are mentioned. A bit of jumping about is needed. If this is a journal- imposed formatting requirement then please ignore.

Reviewer 3 Report

Badarni et al. has shown a nice correlation of HSPD1 and TRAP1 expression with the sensitivity to DNA replication and mitosis inhibitors. 

1. The introduction can be further improved by adding some more background information for the readers 

2. in fig 5A, the quantitation of bands should be done and the blots are missing the protein ladder. Also, a full blot image should be provided in the supplementary. 

3. In fig 5B, the effect of cisplatin has been shown. Did you try any other chemotherapy drugs like Paclitaxel or docetaxel which are also routinely used in treatment of ovarian cancer. It will be interesting to see their effects and correlation because that would help in better understanding about HSPD1. 

4. In fig 5C, usually the cell titre glo assay is shown as % viability. Can you show the results in that format. It will be easier for the readers to understand then and also would be important to see the total number of cells in each sample. 

It is written in the legends of Fig 5C that it is biological replicates from one experiment. However, if it is from one experiment then they are technical replicates. The mean of biological replicates should be shown here rather than technical replicates. 

5. Were the experiments in Fig 5C and 5D also tried in OAW42 cells ? and was the trend similar to that in IGROV1 cells as both these cell lines have higher levels of HSPD1. It will make the conclusions more convincing. 

Author Response

Reviewer #3

Badarni et al. has shown a nice correlation of HSPD1 and TRAP1 expression with the sensitivity to DNA replication and mitosis inhibitors.

  1. The introduction can be further improved by adding some more background information for the readers

Answer: To make the introduction clear, we gave an example of one mitochondrial chaperone, TRAP1, and the protein targets and mitochondrial metabolites that it controls. We also added an explanation about the databases used.

  1. in fig 5A, the quantitation of bands should be done and the blots are missing the protein Also, a full blot image should be provided in the supplementary.

Answer: We thank the reviewer for these comments. We added quantification and protein size to the blots We also added a full blot image in the supplementary.

P.O. Box 653, Beersheva 84105 ISRAEL

  • In fig 5B, the effect of cisplatin has been Did you try any other chemotherapy drugs like Paclitaxel or docetaxel which are also routinely used in treatment of ovarian cancer. It will be interesting to see their effects and correlation because that would help in better understanding about HSPD1.

Answer: We agree that it is interesting that HSPD1 expression is associated with sensitivity to chemotherapy drugs. We were surprised by this finding, as one would expect that a chaperone would be protective. Here, we used cisplatin as a test case and found that, indeed, HSPD1 expression is associated with sensitivity to this drug. While it is interesting to follow up on this finding and test other compounds, in particular ones that are clinically relevant, such experiments are beyond the scope of this work.

  1. In fig 5C, usually the cell titre glo assay is shown as % viability. Can you show the results in that It will be easier for the readers to understand then and also would be important to see the total number of cells in each sample.

Answer: Thank you, corrected.

It is written in the legends of Fig 5C that it is biological replicates from one experiment. However, if it is from one experiment then they are technical replicates. The mean of biological replicates should be shown here rather than technical replicates.

Answer: Thank you, corrected.

  1. Were the experiments in Fig 5C and 5D also tried in OAW42 cells ? and was the trend similar to that in IGROV1 cells as both these cell lines have higher levels of HSPD1. It will make the conclusions more convincing.

Answer: We thank the reviewer for this comment and agree that testing additional cell lines makes this work's conclusion more convincing. Thus, we performed the same experiments in Fig 5C and 5D using OAW42 cells and observed a similar trend. The results for OAW42 cells were added to Figure 5 (Figure 5E and 5F).

Answer: We thank the reviewer for promoting us to discuss other chaperones. We added a paragraph to the Results discussing the point that their expression is associated with drug sensitivity.

  1. The author validated the HSPD1 expression and drug sensitivity It would be better if the authors could also validate TRAP1 expression and drug sensitivity experimentally.

Answer: We agree that testing more chaperones and drugs will make our conclusions stronger. We also agree that this is very interesting. However, these experiments are beyond the scope of the current work. We found interesting correlations between drug sensitivity and the expression of mitochondrial chaperones. This work set the stage for future experiments that we, and hopefully others, will pursue.

  1. The author mentioned that they included the drug category with at least ten drugs The rationale behind this needs to be clarified.

Answer: To infer that a particular category is enriched, we need to have a sufficient number of compounds in the category. We, therefore, removed all categories comprised of fewer than 10 compounds.

  1. In all the bidirectional bar graphs, they mentioned that the black line is a cutoff line for a significant score, which is It should be corrected as a black dotted line. Please also define the X-axis.

Answer: Thank you, corrected.

  1. It is very hard to see the percentage value in all bubble Please enlarge all the bubble plots.

Answer: Thank you, corrected.

Reviewer 4 Report

This is a very good correlative study establishing a link between the mitochondrial chaperon/cochaperone's expression level and the sensitivity of drugs in different cancer types. Although this is not a causality test, it provides an important avenue to investigate further the mechanism behind mitochondrial chaperones’ role in cancer drug sensitivity/resistance. In this manuscript (MS), Badarni et al. performed correlation analyses between the expression level of several genes and different drug categories in many cancer cell lines and tissue subtypes. The authors developed a pipeline to calculate Z-scores and Score-global for several gene-drug combinations in many cell lines. They validated their pipeline with clinically tested drugs against the ErbB2 gene. The authors analyzed a total of 15 mitochondrial chaperons/cochaperones and found that the higher expression of HSPD1 and TRAP1 significantly correlates with sensitivity to replication and mitosis-targeting drugs. I do not think the further analysis is required. However, if the author could address some of the points, it would make this MS look much better and easier to understand.

Minor comments/suggestions

1)    The authors seem to be biased toward drug sensitivity data only. They have skipped explaining the results that show drug resistance with TRAP1 and HSPD1 expression. They have discussed it in the discussion section, but it is better to emphasize both results in the result section. Also, their analysis shows that TRAP1 and HSPD1 are top hits for specific drug sensitivity. However, other chaperon/cochaperones (CLPP, SPG7, HTRA2, AFG3L2) also show significant resistance toward some other drug categories. The authors need to explain and discuss these results.

2)    The author validated the HSPD1 expression and drug sensitivity experimentally. It would be better if the authors could also validate TRAP1 expression and drug sensitivity experimentally.

3)    The author mentioned that they included the drug category with at least ten drugs only. The rationale behind this needs to be clarified.

4)    In all the bidirectional bar graphs, they mentioned that the black line is a cutoff line for a significant score, which is confusing. It should be corrected as a black dotted line. Please also define the X-axis.

           5) It is very hard to see the percentage value in all bubble plots. Please                       enlarge all the bubble plots.

Author Response

This is a very good correlative study establishing a link between the mitochondrial chaperon/cochaperone's expression level and the sensitivity of drugs in different cancer types. Although this is not a causality test, it provides an important avenue to investigate further the mechanism behind mitochondrial chaperones’ role in cancer drug sensitivity/resistance. In this manuscript (MS), Badarni et al. performed correlation analyses between the expression level of several genes and different drug categories in many cancer cell lines and tissue subtypes. The authors developed a pipeline to calculate Z-scores and Score-global for several gene-drug combinations in many cell lines. They validated their pipeline with clinically tested drugs against the ErbB2 gene. The authors analyzed a total of 15 mitochondrial chaperons/cochaperones and found that the higher expression of HSPD1 and TRAP1 significantly correlates with sensitivity to replication and mitosis-targeting drugs. I do not think the further analysis is required. However, if the author could address some of the points, it would make this MS look much better and easier to understand.

Minor comments/suggestions

The authors seem to be biased toward drug sensitivity data only. They have skipped explaining the results that show drug resistance with TRAP1 and HSPD1 expression. They have discussed it in the discussion section, but it is better to emphasize both results in the result section. Also, their analysis shows that TRAP1 and HSPD1 are top hits for specific drug sensitivity. However, other chaperon/cochaperones (CLPP, SPG7, HTRA2, AFG3L2) also show significant resistance toward some other drug categories. The authors need to explain and discuss these results
